# Arthroscopic Treatment of a Subchondral Bone Cyst via Stem Cells Application: A Case Study in Equine Model and Outcomes

**DOI:** 10.3390/biomedicines11123307

**Published:** 2023-12-14

**Authors:** Fernando Canonici, Daniele Marcoccia, Pamela Bonini, Valentina Monteleone, Elisa Innocenzi, Alessia Zepparoni, Annalisa Altigeri, Daniela Caciolo, Silvia Tofani, Paola Ghisellini, Cristina Rando, Eugenia Pechkova, Julietta V. Rau, Roberto Eggenhöffner, Maria Teresa Scicluna, Katia Barbaro

**Affiliations:** 1Equine Practice srl, Campagnano, Strada Valle del Baccano 80, 00063 Roma, Italy; fernandocanonici@gmail.com; 2Istituto Zooprofilattico Sperimentale del Lazio e della Toscana “M. Aleandri”, Via Appia Nuova 1411, 00178 Rome, Italy; daniele.marcoccia@izslt.it (D.M.); pamela.bonini@izsl.it (P.B.); valentina.monteleone-esterno@izslt.it (V.M.); elisa.innocenzi-esterno@izslt.it (E.I.); alessia.zepparoni@izslt.it (A.Z.); annalisa.altigeri@izslt.it (A.A.); daniela.caciolo@izslt.it (D.C.); silvia.tofani@izslt.it (S.T.); teresa.scicluna@izslt.it (M.T.S.); 3Department of Surgical Sciences and Integrated Diagnostics (DISC), Genova University, Corso Europa 30, 16132 Genova, Italy; paola.ghisellini@unige.it (P.G.); cristina.rando@unige.it (C.R.); roberto.eggenhoffner@unige.it (R.E.); 4Consorzio Interuniversitario INBB, Viale delle Medaglie d’Oro, 305, 00136 Roma, Italy; eugenia.pechkova@gmail.com; 5Laboratories of Biophysics and Nanotechnology, Department of Experimental Medicine (DIMES), Genova University, Via Pastore 3, 16132 Genova, Italy; 6Istituto di Struttura della Materia, Consiglio Nazionale delle Ricerche (ISM-CNR), Via del Fosso del Cavaliere 100, 00133 Rome, Italy; giulietta.rau@ism.cnr.it

**Keywords:** subchondral bone cysts, regenerative medicine, horse, adipose tissue mesenchymal stromal cells (AMSCs)

## Abstract

Subchondral bone cysts in horses represent one of the main causes of lameness that can occur in different anatomical locations. The study describes the treatment in regenerative therapy of the intracystic implantation of adipose tissue mesenchymal stromal cells (AMSCs) included in platelet-rich plasma (PRP). The ability of AMSCs to differentiate in osteogenic cells was tested in vitro and in vivo. Given the aim to investigate the application of AMSCs in bone defects and orthopedic pathologies in horses, a four-year-old male thoroughbred racing horse that had never raced before was treated for lameness of the left hind leg caused by a cyst of the medial femoral condyle. The horse underwent a new surgery performed with an arthroscopic approach in which the cystic cavity was filled with AMSCs contained in the PRP. Radiographs were taken 3, 5, and 10 months after the surgery to assess the development of newly regenerated bone tissue in the gap left by the cyst. Twelve months after the operation and after six months of regular daily training, the horse did not show any symptoms of lameness and started a racing career. According to the study, the use of AMSCs and PRP suggests promising benefits for treating subchondral bone cysts.

## 1. Introduction

Subchondral bone cysts (SBCs) in horses represent one of the main causes of lameness and can occur in different anatomical locations [1]. SBC is a multifactorial disease that can be caused by abnormalities in the osteogenesis of the bone epiphyses (osteochondrosis), biomechanical trauma, hormonal and mineral imbalances, bone bruises, diet, and genetic factors. Furthermore, a biochemical etiological component of the disease has also been identified [2], and the osteolytic activity of inflammatory metabolites produced by cystic tissue has been demonstrated. Nevertheless, in horses, SBCs are most commonly found in the medial condyle (MFC) of the femur, followed by phalanges, carpal bones, metacarpal and metatarsal bones, tibia, radius, talus, sesamoid bones, humerus, patella, and tarsal bones [3]. In general, both intermittent lameness and acute lameness can be observed soon after the onset of physical activity [4]. Lameness can vary from mild to severe, and localized clinical signs, such as synovial effusion, may be observed. In 40% of cases, the SBCs communicate with the joint, and in general, bending tests are carried out to determine the affected joint [5]. In horses affected by SBCs, the anatomical localization of the lameness is performed through intra-articular diagnostic anesthesia, followed by other investigations such as radiography (X-ray) [6]. The SBC presents itself radiographically as an isolated subchondral defect that may appear as an ovular, circular, or conical radiolucent shadow with a contour of radiopaque sclerosis, when visible. In horses in which SBC is suspected, it is especially important to radiographically examine the contralateral joint as well, because there may also be bilateral involvement of the affected joints [7]. Nevertheless, in some cases, X-ray alone fails to identify SBC; therefore, other diagnostic tools can be used, such as contrast arthrography and computed tomography (CT) [8], in which the cyst appears as a hypodense area surrounded by a hyperdense area. SBCs, from a histopathological point of view, contain necrotic bone and fibrous tissue surrounding the cyst, leading to the secretion of nitric oxide (NO), prostaglandins (PGs), and neutral metalloproteinases (MMPs), which lead to continuous bone resorption [2,9]. However, SBC is often responsible for relevant clinical manifestations that can lead, for example, to the impairment of normal athletic activity and therefore represents a impairing orthopedic condition that inconsistently responds to treatment [1]. Patients with SBC may benefit from a conservative therapeutic approach, which combines chondroprotective or surgical procedures with the use of medications like corticosteroids or non-steroidal anti-inflammatory drugs (NSAIDs) [10,11]. Different surgical therapeutic approaches can be used, such as arthroscopy or the trans-osseous approach, such as debridement, cystic cavity filling with spongious bone, mosaic arthroscopy, and the grafting of bone replacement materials [12,13]. The prognosis for return to activity ranges from 30% to 80% but depends on the horse’s age, sports activity, the extent of the affected cartilage area, whether or not there is concurrent osteoarthritis, and chosen treatments [14]. This study describes the application of regenerative therapy for the treatment of SBC. The intracystic implantation of adipose tissue mesenchymal stromal cells (AMSCs) included in platelet-rich plasma (PRP), which is a source of growth factors contained in platelet granules such as platelet-derived growth factor (PDGF), insulin-like growth factor (IGF), and tumor necrosis factor (TNF-b); we postulate this treatment could promote and accelerate the healing process and bone recovery. In particular, we focused on evaluating the effectiveness of platelet gel implantation with fat-derived mesenchymal stromal cells for the treatment of the subchondral cyst of the medial condyle of the horse’s femur.

In fact, in reconstructive surgery, the use of platelet gel combined with autologous bone, compared to the use of the latter alone, can increase the speed of bone formation and the density of the newly deposed bone [15,16]. Cell therapy with AMSCs included in PRP could be a valid alternative to traditional therapies for the treatment of SBC. Thus, in the present work, we aim to investigate the application of AMSCs combined with PRP to treat a horse for which two previous surgical approaches failed.

## 2. Materials and Methods

For this study, a four-year-old male thoroughbred racehorse was treated. The horse showed the first clinical signs with lameness at yearling age when he started training, just after he was bought at the yearling sales. The cause of the lameness was related to a subchondral bone cyst of the medial condyle of the left femur. The horse underwent an operation for a cystic enucleation, but after a few months of resting, the lameness relapsed when he restarted the training. Consequently, the horse underwent a second surgery for a second debridement of the cyst, but with the same outcome after four months of rest. In relation to the failure of the previous surgeries, a different treatment was applied, aimed at filling the cyst cavity with AMSCs mixed with PRP gel.

### 2.1. Isolation of Mesenchymal Stromal Cells from Adipose Tissue

AMSCs were obtained from subcutaneous fat collected in the peri-caudal area. Great care was taken during the isolation procedure to ensure sample sterility. The adipose tissue was processed enzymatically using collagenase IA (Sigma, Burlington, MA, USA), which was added to the previously minced adipose tissue at a concentration of 10% *w*/*v*. The mixture was then incubated at 37 °C for 1 h. After 1 h, the sample was centrifuged at 1200× *g* for 8 min, and the pellet was washed three times with αMEM (Gibco, Billings, MT, USA) containing 10% fetal bovine serum (FCS) (Sigma, Burlington, MA, USA). The pellet was then resuspended in αMEM containing 10% FCS and seeded into 75 cm^2^ plastic flasks. The cells were cultured at 37 °C and 5% CO_2_ to facilitate cell proliferation. The medium was changed after two days and then replaced every three days.

Once the cell culture reached 80–90% confluency, the cells were harvested enzymatically using trypsin (0.05%) and EDTA (0.01%) (Gibco, Billings, MT, USA). The cells were then replated, stimulated, and analyzed to evaluate their differentiation potential.

### 2.2. Differentiation of Mesenchymal Stromal Cells

The differentiation potential of AMSCs toward the osteogenic, adipogenic, and chondrogenic lineages was assessed after a second passage of cell growth. In 24-well plates, the cells were seeded at a density of 3 × 10^4^ cells/mL. The appropriate differentiation medium was then used to stimulate the cultures under the following circumstances:

Adipogenic differentiation:

For adipogenic induction, the culture was stimulated for two weeks with αMEM containing 10% FCS supplemented with insulin at a concentration of 10 µg/mL, dexamethasone (Sigma, Burlington, MA, USA) at a concentration of 1 µM, 3-isobutyl-methyl-xanthine (Sigma, Burlington, MA, USA) at a concentration of 0.5 mM, and indomethacin (Sigma, Burlington, MA, USA) at a concentration of 0.2 mM. AMSCs grown in αMEM containing 10% FCS represented the negative control.

The cells were fixed with a 4% formalin solution for 30 min at room temperature after two weeks of differentiation. They were then washed with distilled water and then with 60% isopropanol (Sigma, Burlington, MA, USA) for 5 min. Oil Red O (0.3% in 60% isopropanol) was used to stain the cells for 5 min. Cells were cleaned of stains and then washed in distilled water.

Osteogenic differentiation:

For osteogenic induction, cell culture stimulation was performed for two weeks with a standard medium supplemented with β-glycerophosphate 10 mM (Sigma, Burlington, MA, USA), ascorbic acid 50 µg/mL (Sigma, Burlington, MA, USA), and dexamethasone 10–7 M (Sigma, Burlington, MA, USA). The negative control was represented by AMSCs grown in αMEM containing 10% FCS. After two weeks of osteogenic differentiation, the cells were fixed for 30 min at room temperature in a 4% formalin solution; subsequently, they were washed with distilled water and stained with a solution of Alizarin Red S (2%) in distilled water (Sigma, Burlington, MA, USA) for 30 min. After four washes with distilled water, calcium deposits were highlighted by Alizarin Red S staining.

Chondrogenic differentiation

For chondrogenic induction, cell culture stimulation was performed for two weeks with αMEM containing 1% FCS supplemented with insulin 6.25 µg/mL (Sigma, Burlington, MA, USA), dexamethasone 0.1 µM (Sigma, Burlington, MA, USA), TGF-beta3 10 ng/mL (Sigma, Burlington, MA, USA), and ascorbic acid 50 nM (Sigma, Burlington, MA, USA). AMSCs grown in αMEM containing 10% FCS served as the negative control.

The cells were fixed with a 4% formalin solution for 30 min at room temperature after two weeks of chondrogenic differentiation. They were then washed with distilled water, stained with Alcian Blue in Acetic Acid (Sigma, Burlington, MA, USA) (1% in a 3% solution of acetic acid; pH 2.5) for 15 min at room temperature, and washed three times with 3% acetic acid.

The cells were observed using an inverted microscope (Nikon ECLIPSE TE2000-U; Nikon Minato, Tokyo, Japan), and images were captured with a camera (Nikon Digital Sight 10; Nikon Minato, Tokyo, Japan).

### 2.3. Preparation of Platelet Gel Containing AMSCs for the Treatment of Bone Cyst

The horse’s venous blood was taken at the time of surgery (54 mL). Sodium citrate was added in the amount of 6 mL at this volume to stop coagulation. The sample of blood was centrifuged at 800× *g* for 15 min at 20 °C to divide it into two phases. The selected acceleration causes the blood to separate into two distinct phases: an upper portion of plasma containing platelets and a lower portion of sedimented red blood cells. The plasma was once again centrifuged at 1200× *g* for 10 min to sediment the platelets.

The sediment was resuspended in the remaining volume after approximately one-third of the volume was discarded, yielding approximately 20 mL of platelet-rich plasma (3–4 times the baseline value).

The AMSCs at passage 2 of cell growth were enzymatically harvested using Trypsin-EDTA (total of 4,800,000 cells) (Sigma, Burlington, MA, USA). The pellet was washed three times with PBS and added to platelet-rich plasma. Finally, 10% calcium chloride (100 µL/1 mL plasma) was added to obtain the coagulation of the fibrin. This process resulted in the creation of a gel that contained both autologous platelets and AMSCs and was prepared for quick injection into the injured anatomical site (within 20 min of the completion of coagulation).

### 2.4. Surgical Treatment of the Clinical Case

The horse underwent anesthesia with premedication of 300 mg of xylazine, 500 mL of glyceric ether guaiacol (7.5%), and 1 g of intravenous ketamine hydrochloride to induce the anesthesia that was maintained with oxygen and isoflurane administered through endotracheal intubation. The horse was positioned in dorsal recumbency, with the hind limb supported to flex the femoral tibial–patellar joint at approximately 90 degrees.

Just after trichotomy, the knee joint area was thoroughly cleaned and subjected to numerous cycles of application of chlorhexidine and alcohol before the surgical field was covered with sterile adhesive drapes and numerous sterile drapes to isolate the knee area.

The arthroscope sheath was inserted using a conventional lateral approach. The joint was distended with sterile sodium chloride solution injected into the joint with an arthroscopic pump, which also assured the maintenance of constant intra-articular pressure. A second surgical access was precisely directed over the cystic cavity after highlighting the cartilaginous defect that corresponded to the entrance of the cyst cavity. A small opening was made on the surface of the joint by debriding the superficial fibrocartilage and bone layers to insert an arthroscopic lavage cannula to flush the joint first, then to inject the cyst with the use of a 20 mL syringe previously filled with platelet-rich plasma containing AMSCs after having distended the joint with carbon dioxide gas. The cystic cavity was filled and compacted by applying pressure with an arthroscopic probe. The two small arthroscopic incisions were sutured in a simple, interrupted pattern.

The surgery was carried out using a 4 mm Storz arthroscope (KARL STORZ SE & Co. KG Tuttlingen, Germany) with a 30° viewing angle connected to a Sony (Sony Corporation, Konan Minato-ku, Tokyo, Japan). Monitor through a Storz medical camera (KARL STORZ SE & Co. KG Tuttlingen, Germany). The intraoperative phases were memorized through the acquisition of images with a Medicapture device (Plymouth Meeting, PA, USA).

### 2.5. Radiographic Analysis

Following sedation, a radiographic examination of the femoral–tibial joint was performed with a caudo-cranial view to evaluate the femoral condyles and, consequently, the cyst area. This was carried out at three, five, and ten months following surgery to monitor the evolution of the bone void of the cyst.

## 3. Results

The micrograph in Figure 1 represents the shape and arrangement of mesenchymal stromal cells (MSCs) isolated from adipose tissue at the second passage after isolation, as discussed in earlier sections. These cells are elongated and spindle-shaped, with a large nucleus and thin cytoplasm. They are arranged in a random pattern on the culture dish, and the details of their structure and the spaces between them are discernible. Their morphology closely resembles micrographs of mesenchymal stromal cells from horse adipose tissue [17].

The horse AMSCs were examined for their capacity to differentiate into osteogenic, chondrogenic, and adipogenic lineages using the procedures outlined in the Materials and Methods. In Figure 2, mesenchymal stromal cells subjected to osteogenic, adipogenic, and chondrogenic culture conditions are shown to differentiate after two weeks in culture.

In Figure 2a,b, the results of osteogenic differentiation of equine mesenchymal stromal cells after two weeks of culture in an osteogenic medium are reported. Figure 2a is the negative control, where AMSCs were cultured in a basal medium without osteogenic factors. Figure 2b shows the results of osteogenic differentiation. The image on the right (b) shows a higher intensity of Alizarin Red S staining, which binds to calcium salts, a major component of the bone matrix, indicating a higher production of calcium salts by the differentiated osteoblasts.

Figure 2c,d show the chondrogenic differentiation. Figure 2d shows the intensity of Alcian Blue staining, indicating the production of glycosaminoglycans by the differentiated chondrocytes and the diffused presence of components of the cartilage matrix.

Figure 2e,f show the results of adipogenic differentiation in an adipogenic medium. Figure 2e is the negative control, where AMSCs were cultured in a basal medium without adipogenic factors. Figure 2f shows a higher intensity of Oil Red O staining, indicating a higher accumulation of lipid droplets by the differentiated adipocytes.

Figure 2f show adipose differentiation, and the enlargements in Figure 2g,h exhibit clusters of cells that have undergone adipogenic differentiation of AMSCs. The cells are elongated and have a red-orange color due to the staining of lipid droplets with Oil Red O. The cells appear to be in close contact and are overlapping each other, suggesting a high density of adipocytes. Some of the cells’ nuclei are visible due to the cells’ translucent nature.

Globally speaking, the findings in Figure 2 imply that osteogenic, chondrogenic, and adipogenic differentiations of AMSCs can be induced by using a particular medium.

Additionally, the staining compounds used for the various lineages represent a trustworthy method to assess the differentiation potential of AMSCs.

### 3.1. Bone Cyst Treatment

Figure 3 represents an arthroscopic intraoperative image of the cyst already filled with a gel, containing autologous platelets and autologous AMSCs, that was directly injected into the cyst cavity after its debridement.

The subchondral cyst is radiographically visible in the medial femoral condyle of the left stifle joint in a caudo-cranial view (Figure 4a) and measures approximately 2 cm in diameter. The outline of the medial femoral condyle on the joint surface is flattened.

### 3.2. Clinical Evaluation

Following the arthroscopic procedure, the horse was discharged from the hospital a week later with indications of one month of rest in the stall, then started a progressive walking exercise for up to four months, then started a ridden progressive exercise to resume normal training for a thoroughbred racehorse. The cyst evolution was radiographically monitored at 3 (Figure 4b), 5 (Figure 4c), and 10 months (Figure 4d). The radiolucency of the cyst progressively reduced and finally became very light and smaller in size.

The follow-up radiographs demonstrate the almost complete healing of this chronic and unresponsive condition, which is also supported by the favorable clinical response to the treatment.

## 4. Discussion

Mesenchymal stromal cells can differentiate into a variety of mesenchymal lineages, including osteoblasts, chondrocytes, adipocytes, and myocytes when they are isolated from adipose tissue [18]. As reported, after a few weeks of stimulation, the equine AMSCs underwent multi-lineage differentiation after treating the cultures with osteogenic, chondrogenic, and adipogenic inducers [19]. The amount of sampling is also greatly reduced because the stromal cells isolated from this alternative source have exceptionally high proliferative capacities. The combination of these characteristics makes adipose tissue an excellent source of mesenchymal progenitor cells, meeting the characteristics necessary for their use in regenerative medicine [20,21]. The regenerative potential of the bone may not be sufficient to guarantee effective “*restitutio ad integrum*” [22,23]. The potential for self-repair of bone tissue is based on the existence of compartmental stromal cells. However, there are limits to the ability to repair bone tissue; they are primarily related to the amount of tissue that needs to be reconstructed, the space between the lesion’s edges, and the amount of time required for the reconstruction. Thus, the AMSCs were combined with the PRP gel.

It is crucial to remember that medial femoral condyle cysts, one of the developmental orthopedic diseases in horses, can show symptoms even at a young age. In fact, a novel method utilizing absorbable screws implanted transcortically into the cyst that filled the bone void has been proposed to achieve the resolution of the lameness [24] without leaving an on-site implant like a conventional cortical screw, as suggested previously [25] and more recently [26], with the latter method yielding a less favorable result. The ideal outcome of the cystic lesion treatment would be addressed to the resolution of clinical symptoms as well as to achieve the anatomic recovery of the bone defect in absence of the implant detectable on radiographs.

This study investigated the application of adipose mesenchymal stem cells with PRP gel for bone regeneration in a horse affected by SBC of the left medial femoral condyle. Harvesting adipose tissue from horses is an easy procedure which offers a rich source of MSC. They also showed a typically high proliferative capacity, a property that allows for minimizing the amount of sampling required. In the laboratory, MSCs have also shown multipotency in differentiating into osteogenic, chondrogenic, and adipogenic lineages. In the study, MSCs already incorporated into the platelet gel were implanted arthroscopically into the bone void of the cyst to suggest a novel method for treating subchondral bone cysts in horses that is different than that proposed for treating subchondral bone cysts of the femoral condyle unresponsive to the traditional treatments [27] and that more recently proposed by direct multiple injections into the bone cyst.

For this purpose, a 4-year-old thoroughbred racehorse was treated in our clinical case. This horse had previously undergone two arthroscopic procedures to enucleate and remove the SBC tissue, but radiographic analyses revealed no improvement of the pathology nor resolution of the lameness. As a result, it underwent a new surgery that was carried out using a traditional arthroscopic approach, through which the cystic cavity was then filled with AMSCs that were included in PRP gel.

Three months after surgery, a decrease in the cyst size and an increase in subchondral plate thickness were seen radiographically. This improvement was even more obvious after five months; in fact, there was a 50% reduction in the cystic cavity and a significant increase in the subchondral plate. The horse showed no recurrence of the lameness after 12 months following the operation and after six months of normal training. Moreover, the horse, which had never raced before the surgery, was able to race at 5 years old with good performance. The clinical case is significant as it suggests the favorable effect of AMSCs and PRP in the regeneration of bone tissue of a subchondral bone cyst and motivates further research to truly demonstrate its effect.

Given the outcome of the clinical case described, even with the limitation of the procedure being performed in only one case, this approach can be seriously considered for the treatment of SBCs in horses, with the potential to achieve excellent results. It is crucial to note that the horse had previously undergone two traditional arthroscopic procedures that had failed, and the horse substantially remained lame for two years. Further, other treatments carried out with stem cells without the addition of PRP have also shown limited effectiveness compared to the results of the present treatment [28].

Possible developments from the results of the present research could concern, for example, three-dimensional biomaterials that combine AMSCs with biomimetic scaffolds composed of natural or synthetic materials. These biomaterials can promote their adhesion, growth, proliferation, and differentiation. The development of novel biomaterials or scaffolds that might enhance the delivery and retention of AMSCs at the target site could improve their survival and differentiation in vivo. Furthermore, future research might explore the immunomodulatory properties of AMSCs and their role in modulating the inflammatory response and preventing rejection after transplantation.

## 5. Conclusions

The clinical case of a horse with a subchondral cyst of the medial condyle of the femur was treated in the present study, which focused on applications of MSCs from adipose tissue to heal horse bone cyst conditions. The platelet gel, a blood component with growth factors favoring clotting and healing, was combined with the MSCs. A minimally invasive arthroscopy surgical procedure was used to implant the MSCs in platelet gel into the cyst cavity. The horse made a full recovery from the operation and was able to resume training after five months and start a racing career, with only one episode of severe lameness. Also, the radiographs revealed that the treated cystic cavity gradually improved through bone tissue regeneration.

The proposed treatment of this study can be considered for the management of this fairly common horse condition that has been treated with a variety of modalities, some of which were able to improve the clinical signs without resolving the cyst’s bone void. The failure to solve the bone defects represents a significant restriction on the outcome of having the bone free of the cyst, as detectable on radiographs. This has an important impact on the horse industry as it limits the saleability of a horse and subsequently causing significant losses to horse breeders.

To achieve greater scientific validation, a larger number of horses should be treated using the methodology proposed in this case, and a control group should be used study to evaluate the response to the treatments with either AMSCs alone or with PRP gel alone, considering that MSCs injected into the cyst void did not show different results compared to the corticosteroid injection and cyst debridement [28]. Future studies could also investigate the effectiveness of AMSCs trapped in a fibrin network in a larger sample of horses with subchondral bone cysts or explore their potential use in treating other orthopedic pathologies.

## Figures and Tables

**Figure 1 biomedicines-11-03307-f001:**
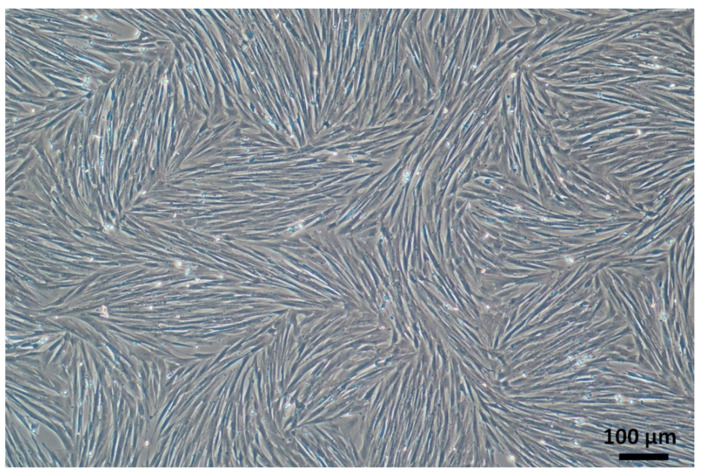
Micrograph of MSCs at the second passage after isolation from the adipose tissue of the horse.

**Figure 2 biomedicines-11-03307-f002:**
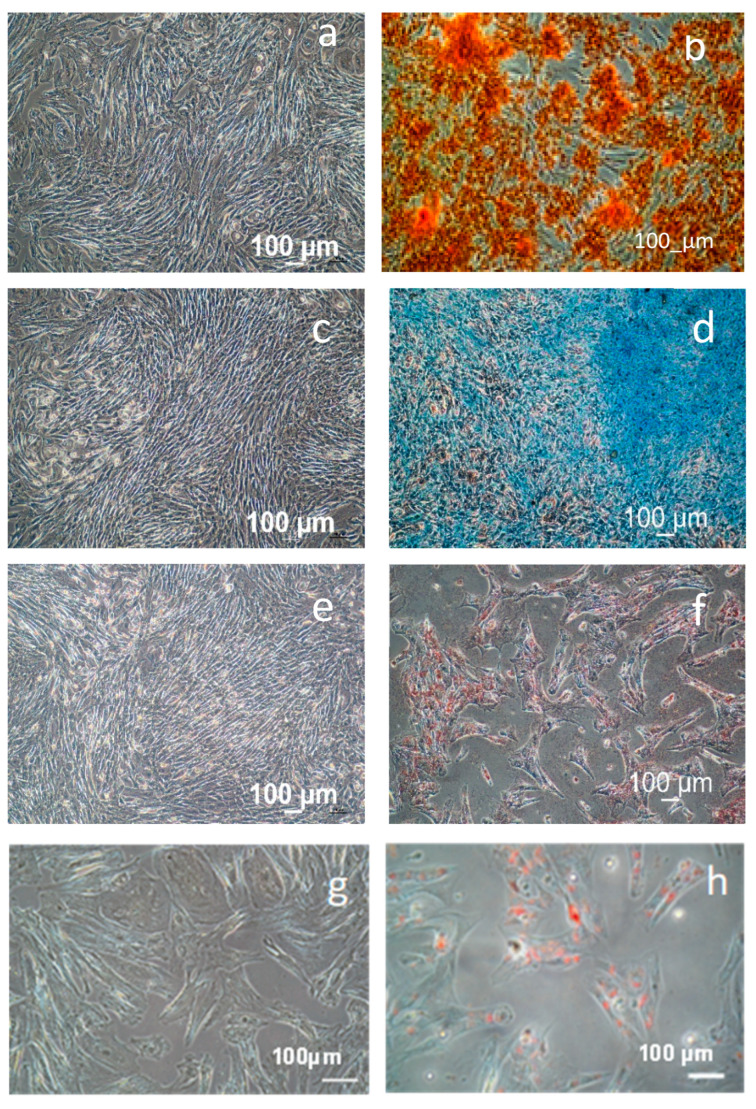
Differentiation of cells after two weeks of treatment, as detailed in Section 2. (**a**,**b**) depict osteogenic differentiation, (**c**,**d**) illustrate chondrogenic differentiation, and images (**e**–**g**) and h demonstrate adipogenic differentiation. Images (**a**,**c**,**e**) serve as negative controls, representing AMSCs cultured solely in αMEM/10% FCS. The differentiated cells are depicted in images (**b**,**d**,**f**–**h**).

**Figure 3 biomedicines-11-03307-f003:**
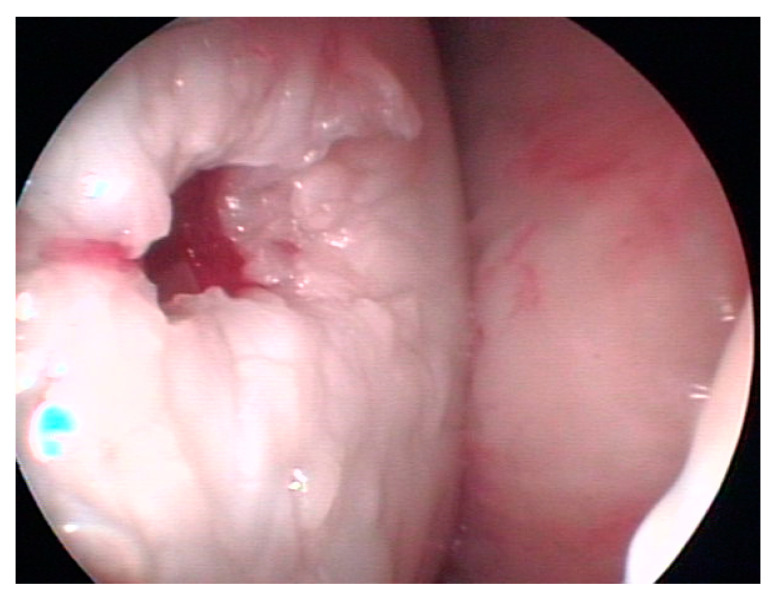
Arthroscopic view of the filling of the cystic cavity with platelet gel.

**Figure 4 biomedicines-11-03307-f004:**
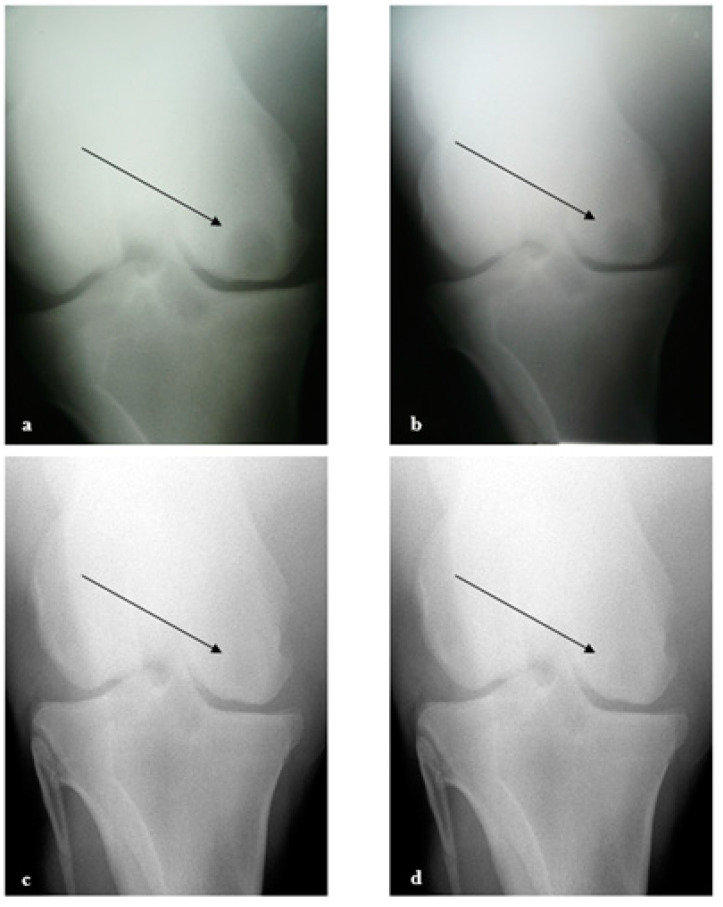
Radiographic evolution of the subchondral cyst (**a**) preoperative situation; (**b**) after 3 months; (**c**) after 5 months; and (**d**) 10 months after the operation. Arrows indicate the region of the subchondral cyst.

## Data Availability

All the relevant data are reported in the paper.

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
