# Peer review of "Arthroscopic Treatment of a Subchondral Bone Cyst via Stem Cells Application: A Case Study in Equine Model and Outcomes"

_biomedicines, 2023, doi:10.3390/biomedicines11123307_

Round 1
Reviewer 1 Report (Previous Reviewer 1)
Comments and Suggestions for Authors
General Comments
This is a limited clinical study where a subchondral bone cysts in a single horse joint was repaired using adipose derived MSCS mixed with platelet rich plasma. The manuscript is presented as an article, but it is at best a clinical report and should be presented as such. There is no experimental study presented and the previous history of the horse was not presented so that it is difficult to compare the progression of healing from previous work to the current work.
Specific Comments
Figure 1. The magnification of this photo is not 10x!! Please provide an accurate magnification. Also, the passage number of the cells should be included in the legend. The morphologic profile of these cells may be homogeneous, but this does not mean that the cellular population is physiologically homogeneous.
Placement of the cursor over Figure 3 indicates that this a photo of human tissue. If this is accurate it is inappropriate for this manuscript.
Author Response
General Comments
This is a limited clinical study where a subchondral bone cysts in a single horse joint was repaired using adipose derived MSCS mixed with platelet rich plasma. The manuscript is presented as an article, but it is at best a clinical report and should be presented as such. There is no experimental study presented and the previous history of the horse was not presented so that it is difficult to compare the progression of healing from previous work to the current work.
We have accepted your observation and have changed the title although we prefer that it be accepted as work.
Specific Comments
Figure 1. The magnification of this photo is not 10x!! Please provide an accurate magnification. Also, the passage number of the cells should be included in the legend. The morphologic profile of these cells may be homogeneous, but this does not mean that the cellular population is physiologically homogeneous.
We have modified as from your indications by inserting the actual scale bar with the unit of measurement which makes the reference to the enlargement factor superfluous in both Figure 1 and Figure 2.
Placement of the cursor over Figure 3 indicates that this a photo of human tissue. If this is accurate it is inappropriate for this manuscript.
We have eliminated the option of automatically generating text for images from the Word settings, which in this specific case inserted automatically an incorrect caption that doesn’t correspond to reality. Further it doesn’t enter in the editorial process. We thank the reviewer for having point out these hidden annotation.
Reviewer 2 Report (Previous Reviewer 2)
Comments and Suggestions for Authors
Dear authors,
I agree that your case report is of interest. However, I think that the the rigor of the paper should be further improved.
E.g.: A treatment that was tried once and is reported in a case report should not be addressed as "established treatment method".
Thank you very much for removing the sentences “The regenerative therapy method combined with the minimally invasive technique can offer an effective alternative for the treatment of other orthopedic pathologies in both animals and humans” and “The use of AMSCs with osteogenic potential, combined with platelet gel, could be the optimal solution for the repair of non-critical dimension bone defects” to avoid missleading.
It is acknowledged and understood that there are differences between the work of Klein et al. 2022 and your work. However, since Klein et al. 2022 showed (as you also highlight in your response) that treatment with a) MSCs with b) arthroscopic debridement and c) corticosteroids showed no differences in recovering limb functionallity or production of new bone tissue, there is lack of an indication that MSCs would have a superior treatment effect and that they are required to achieve improvement. Due to the lack of a control group in the current work there is hence no proof, that the reported treatment effect was achieved by the combination of PRP-Gel and AMSCs. Actually the effect could be based merely on the PRP Gel allone which contains a lot of pro-regenerative growthfactors (as outlined by the authors).
A statement should be included to explain that the reported effect could have resulted from the AMSCS allone, the PRP allone or the combination of both and that the question what ultimeately led to the positive effect can not be answered and will require further investiation. Since in other studies debridement and corticosteroids also showed some effect, future studies should not only include more horses but also controlgroups offering a comparison to these treatments . Please discuss this in your manuscript and indicate what would be required in future studies in order to show if the treatment effect was truely based on the combination of PRP and AMSCs or either one of them allone.
Wrong citation Lines 80-83:
The intracystic implantation of adipose tissue mesenchymal stromal cells (AMSCs) included in platelet-rich plasma (PRP), which is a source of growth factors contained in platelet granules such as platelet-derived growth factor (PDGF), insulin-like growth factor (IGF), and tumor necrosis factor (TNF-b), can promote and accelerate the healing process and bone recovery [17].
Nardini et al. 2020 never claimed that intracystic implantation of adipose tissue mesenchymal stromal can promote and accelerate bone recovery. Please rephrase.
Line 36 please remove the part of the sentence which was removed from the last sentence of the introduction.
Comments on the Quality of English LanguageThe English still needs extensive editing. Please find a few examples below. However, the list is not exhaustive. The examples were chosen to illustrate the English mistakes, but there are many more throughout the manuscript. Please ask a native speaker to edit the manuscript
Abstract., Lines 33-35:
According to the study, the use of AMSCs and PRP emerges as an established method for treating subchondral bone cysts in horses suggests a promising benefit of using AMSC and PRP for treating subchondral bone cysts.
Lines 71-72:
The therapeutic approach used to treat SBC can be of type conservative, which involves the use of drugs such as corticosteroids or non-steroidal anti-inflammatory drugs (NSAIDs) in association with chondroprotective or surgical type
M&Ms
...a four-year-old male English thoroughbred employed in racing gallops
Did you meen gallop racing?
Author Response
Dear authors,
I agree that your case report is of interest. However, I think that the the rigor of the paper should be further improved.
E.g.: A treatment that was tried once and is reported in a case report should not be addressed as "established treatment method".
The typo has been eliminated.
Thank you very much for removing the sentences “The regenerative therapy method combined with the minimally invasive technique can offer an effective alternative for the treatment of other orthopedic pathologies in both animals and humans” and “The use of AMSCs with osteogenic potential, combined with platelet gel, could be the optimal solution for the repair of non-critical dimension bone defects” to avoid misleading.
It is acknowledged and understood that there are differences between the work of Klein et al. 2022 and your work. However, since Klein et al. 2022 showed (as you also highlight in your response) that treatment with a) MSCs with b) arthroscopic debridement and c) corticosteroids showed no differences in recovering limb functionality or production of new bone tissue, there is lack of an indication that MSCs would have a superior treatment effect and that they are required to achieve improvement. Due to the lack of a control group in the current work there is hence no proof, that the reported treatment effect was achieved by the combination of PRP-Gel and AMSCs. Actually, the effect could be based merely on the PRP Gel alone which contains a lot of pro-regenerative growth factors (as outlined by the authors).
A statement should be included to explain that the reported effect could have resulted from the AMSCS alone, the PRP alone or the combination of both and that the question what ultimately led to the positive effect cannot be answered and will require further investigation. Since in other studies debridement and corticosteroids also showed some effect, future studies should not only include more horses but also control groups offering a comparison to these treatments. Please discuss this in your manuscript and indicate what would be required in future studies in order to show if the treatment effect was truly based on the combination of PRP and AMSCs or either one of them alone.
Regarding the study of the effect of the combination of PRP and AMSC we have included the correct reference [17] Adipose tissue-derived mesenchymal stem cells and platelet-rich plasma: stem cell transplantation methods that enhance stemness. doi:10.1186/s13287-015-0217-8
Wrong citation Lines 80-83:
The intracystic implantation of adipose tissue mesenchymal stromal cells (AMSCs) included in platelet-rich plasma (PRP), which is a source of growth factors contained in platelet granules such as platelet-derived growth factor (PDGF), insulin-like growth factor (IGF), and tumor necrosis factor (TNF-b), can promote and accelerate the healing process and bone recovery [17].Nardini et al. 2020 never claimed that intracystic implantation of adipose tissue mesenchymal stromal can promote and accelerate bone recovery. Please rephrase.
The reference has been eliminated and the sentence has also been replaced with one consistent with the discussion on the work.
Line 36 please remove the part of the sentence which was removed from the last sentence of the introduction.
We thank the reviewer for having pointed out our typo error that we removed.
The English still needs extensive editing. Please find a few examples below. However, the list is not exhaustive. The examples were chosen to illustrate the English mistakes, but there are many more throughout the manuscript. Please ask a native speaker to edit the manuscript.
Abstract., Lines 33-35: According to the study, the use of AMSCs and PRP emerges as an established method for treating subchondral bone cysts in horses suggests a promising benefit of using AMSC and PRP for treating subchondral bone cysts.
Lines 71-72:The therapeutic approach used to treat SBC can be of type conservative, which involves the use of drugs such as corticosteroids or non-steroidal anti-inflammatory drugs (NSAIDs) in association with chondroprotective or surgical type
M&Ms ...a four-year-old male English thoroughbred employed in racing gallops.
Did you mean gallop racing?
The work has been revised the text we have decided to ask a final revision by a native speaker biologist as suggested by Biomedicines.
Round 2
Reviewer 1 Report (Previous Reviewer 1)
Comments and Suggestions for Authors
The authors have finally!!! made so significant corrections.
Author Response
We thank the reviewer for his/her contribution to the present paper.
Reviewer 2 Report (Previous Reviewer 2)
Comments and Suggestions for Authors
Thank you very much for implementing the requested changes.
There are a few minor things left to be rephrased or removed.
Line 161-163: "High rotational speeds cause the blood to separate into two distinct phases: an upper portion composed of platelet-rich plasma and a lower portion of sedimented red blood cells."
The plasma layer (portion) obtained after a first centrifugation for seperating the red blood cells from the plasma fraction is considered platelet poor plasma not platelet rich plasma. The platelets are only enriched in the second centrifugation step. Please rephrase.
Line 288-291: "In particular, the use of autologous AMSCs opens up a wide range of potential applications, primarily avoiding the rejection-related problems [22] that can also cause animal death and the avoidance of immunosuppressive therapies that, over time, would weaken the immune system and could also determine various types of cancer."
This is an outdated concern or information for the treatment of humans. It has by now been shown that allogeneic MSC treatments in animals are safe, do not cause rejection (at least not upon first application and after local injection) and do not require immunosuppressive therapies. There are even commercially available EMA certified allogeneic MSC based products on the market by now (e.g. Arti-Cell® FORTE or RenuTend®). Please remove or rephrase this statement about the risk of allogeneic treatments.
Line 313-315: "The clinical case is significant as it demonstrates the effectiveness of AMSCs and PRP in the regeneration of bone tissue of a subchondral cyst..."
A case report with n=1 can not demonstrate effectiveness. It can maybe indicate a potential beneficial effect and thereby trigger/initiate/warrent further research efforts in order to truely demonstrate an effect. Please rephrase.
Conclusion section:
A conclusion should offer the main take home message from the paper in a few sentences and not a summary of the whole paper. Please move important information to the discussion and remove repetitive info and formulate a condensed take home message for the conclusion section.
Comments on the Quality of English LanguageThanks for corracting the English. A few minor things may still be changed but can be corrected upon working on the proofs.
Author Response
There are a few minor things left to be rephrased or removed.
Line 161-163: "High rotational speeds cause the blood to separate into two distinct phases: an upper portion composed of platelet-rich plasma and a lower portion of sedimented red blood cells."
The plasma layer (portion) obtained after a first centrifugation for seperating the red blood cells from the plasma fraction is considered platelet poor plasma not platelet rich plasma. The platelets are only enriched in the second centrifugation step. Please rephrase.
We accept the reviewer's suggestion and reworded it in red.
Line 288-291: "In particular, the use of autologous AMSCs opens up a wide range of potential applications, primarily avoiding the rejection-related problems [22] that can also cause animal death and the avoidance of immunosuppressive therapies that, over time, would weaken the immune system and could also determine various types of cancer."
This is an outdated concern or information for the treatment of humans. It has by now been shown that allogeneic MSC treatments in animals are safe, do not cause rejection (at least not upon first application and after local injection) and do not require immunosuppressive therapies. There are even commercially available EMA certified allogeneic MSC based products on the market by now (e.g. Arti-Cell® FORTE or RenuTend®). Please remove or rephrase this statement about the risk of allogeneic treatments.
We accept the reviewer's suggestion and remove the full statement.
Line 313-315: "The clinical case is significant as it demonstrates the effectiveness of AMSCs and PRP in the regeneration of bone tissue of a subchondral cyst..."
A case report with n=1 can not demonstrate effectiveness. It can maybe indicate a potential beneficial effect and thereby trigger/initiate/warrent further research efforts in order to truely demonstrate an effect. Please rephrase.
We accept the reviewer's suggestion and reworded it in red.
Conclusion section:
A conclusion should offer the main take home message from the paper in a few sentences and not a summary of the whole paper. Please move important information to the discussion and remove repetitive info and formulate a condensed take home message for the conclusion section.
We have eliminated repetitions due to a typo and reorganized discussions and conclusions as required. We thank the reviewer for having point out this mistake.
Thanks for corracting the English. A few minor things may still be changed but can be corrected upon working on the proofs.
We have agreed with the publisher to foresee a final revision by a native English biologist and we would maintain our commitment.
This manuscript is a resubmission of an earlier submission. The following is a list of the peer review reports and author responses from that submission.
Round 1
Reviewer 1 Report
Comments and Suggestions for Authors
General Comments
This is a limited clinical study where subchondral bone cysts in a single horse joint was repaired using adipose derived MSCS mixed with platelet rich plasma.
IRB approval is not required since no human subjects or cells were employed in the study. However, approval is required for the employment of a vertebrate animal and cells from such an animal. Also, the owner of the horse must supply approval. These approvals must be presented to make this study acceptable for publication.
Specific Comments
Figure 1. The magnification is not 10x. Also, the passage number of the cells should be included in the legend. The morphologic profile of these cells may be homogeneous, but this does not mean that the cellular population is physiologically homogeneous.
L213 and elsewhere; stem cells, please do not use the term stem cells as this has been disproven in many different studies. These cells do possess therapeutic potential and a possible alternate term could be therapeutic cells. Also, the authors have provided no in vivo evidence that the applied cells differentiated as chondrocytes or osteocytes.
Image 3. Is this the image of the horse surgery? The image says that it is of a human subject.
Author Response
We thank you for the careful analysis of the study and we attach the document certifying the approval by the owner of the horse.

Reviewer 2 Report
Comments and Suggestions for Authors In the submitted manuscript the authors present the case (n=1) of a thoroughbred racehorse with a subchondral bone cyst (SBC) which was treated with adipose derived MSC in a platelet rich gel. Although, it is really remarkable that the horse responded positively to the MSC+PRP-Gel treatment after failure of 2 other surgeries using a different technique, the novelty of the approach is low. MSCs have been imployed for the treatment of subchondral bone cysts for a long time and even large studies (e.g. Klein et al. 2022) including a total of 107 thoroughbreds of which 19 horses had been treated with an intra-cystic application of MSCs have been carried out. The study by Klein et al. showed no significant differences in the ability of unraced thoroughbreds to race after treatment of SBC (in the same anatomic location as in the current manuscript) with either arthroscopic debridement, intralesional mesenchymal stem cells, or intralesional corticosteroids. So the onyl novelty of the presented manuscript is the inclusion of the PRP-gel. However, also in the study by Klein et al. 2022 84% of the horses treated with MSCs returned to racing, so there is no evidence that the inclusion of the PRP-Gel made a difference or not, since the presented study was carried out in n=1. Therefore, the conclusions drawn in the manuscript are clearly not supported by the results: The use of AMSC and PRP does not "emerge as an established method for treating subchondral bone cysts in horses" and the manuscript can not offer evidence of an effective treatment particularaly not for "the treatment of other orthopedic pathologies" (stated but not analysed within the scope of the study) and particularly not "in both animals and humans" (stated but not analysed within the scope of the study). Everything else presented in the manuscript (isolation and trilinage differentiation of adipose derived MSC) is not new but established routine. Therefore unfortunately the manuscript can not be recommended for publication but should be rejected.Vet Surg. 2022 Apr;51(3):455-463. doi: 10.1111/vsu.13782
Comparative results of 3 treatments for medial femoral condyle subchondral cystic lesions in Thoroughbred racehorses
Chelsea E Klein, Lawrence R Bramlage, Darko Stefanovski, Alan J Ruggles, Rolf M Embertson, Scott A Hopper
Comments on the Quality of English LanguageUnfortunately the quality of the English language is poor. Would recommend asking a native English speaker to read and correct future articles prior to submission.
Author Response
We thank you for your efforts and valuable comments. We complied with all requests, answering the questions they raised. The most substantial changes have been highlighted in red in the text, the English has been corrected. We have examined Klein's article and included in our work a comparative discussion and related reference implicit in his comment.
Reviewer 3 Report
Comments and Suggestions for Authors
This is an excellent study that brings a lot of well-studied results to basic researchers and clinicians. Still unique but highly expected study on bone cyst treatment by stem cells in horses. I did not find any weaknesses.
Author Response
We thank you for the careful analysis of our study and the appreciation in the review.
Reviewer 4 Report
Comments and Suggestions for Authors
Comments to the Authors of manuscript ID: biomedicines-2616454 entitled “Arthroscopic Treatment of a Subchondral Bone Cyst by Stem Cells Application: A Case Study in Equine Model and Outcomes”.
This study treated a horse's subchondral bone cyst with adipose tissue mesenchymal stem cells (AMSC) in platelet-rich plasma (PRP). The horse recovered after surgery and showed no lameness, indicating that this approach is effective for treating such cysts in horses.
1. The text contains several grammatical and stylistic issues that could be improved for clarity.
2. The introduction provides essential background information on subchondral bone cysts (SBC) in horses but does not explicitly state a hypothesis or the aim of the study. Typically, a hypothesis or the aim is presented after the background information to guide the reader on what the study aims to achieve. This introduction sets the stage for discussing the treatment using adipose tissue mesenchymal stem cells (AMSC) in platelet-rich plasma (PRP) but does not provide a clear research question or objective.
3. The choice of the four-year-old male English Thoroughbred racing horse for the study appears to be based on a specific clinical history. The horse had a pre-existing subchondral bone cyst (SBC) in the medial condyle of the right femur, which was initially diagnosed when he was 18 months old in Newmarket, England. After experiencing lameness, the horse underwent two prior surgeries for cystic enucleation and debridement, both of which did not provide a satisfactory outcome.
Given this clinical history, the decision to select this horse for a different treatment involving the intra-cystic implantation of adipose tissue mesenchymal stem cells (AMSC) mixed with platelet-rich plasma (PRP) may be reasonable. This choice allows the researchers to investigate the potential effectiveness of this new treatment approach in a case where previous treatments had not been successful.
However, the appropriateness of this choice would depend on the specific research goals of the study, and it's important to consider factors such as the horse's age, clinical condition, and previous treatments in the study design.
4. The procedure described for isolating mesenchymal stem cells from adipose tissue (AMSCs) appears to be thorough and well-documented. It includes the use of appropriate reagents and protocols for maintaining sterility during the isolation process. The subsequent differentiation of these cells into osteogenic, adipogenic, and chondrogenic lineages is clearly described.
The preparation of the platelet gel containing AMSCs for treatment is also well-explained, detailing the steps involved in obtaining platelet-rich plasma and the mixing of AMSCs with it to create the gel.
The surgical treatment of the clinical case is described in a step-by-step manner, providing details on anesthesia, surgical access, and the application of the gel within the cystic cavity.
The radiographic analysis is briefly outlined, stating that radiographic examinations were performed at three, five, and ten months following the surgery to assess the evolution of the bone void within the cyst.
Overall, the procedures and methods are clearly presented, and the descriptions are sufficiently detailed to understand the steps involved in the study.
5. The description of the results and figures in the provided text appears to be clear and informative. However, there are some grammatical issues and awkward phrasing that could be improved for clarity.
6. there are a few improvements that can enhance the discussion section meritoriously:
Interpretation of Specific Findings: The discussion could benefit from a more direct interpretation of the specific results presented earlier in the text. For example, the discussion might relate the successful differentiation of AMSC into various lineages to the potential of these cells in regenerative therapy.
Clinical Significance: While the text mentions the clinical case involving the English Thoroughbred horse, a more comprehensive discussion of the clinical outcomes, their significance, and the potential impact on the treatment of subchondral bone cysts in horses would strengthen the discussion.
Addressing Limitations: It's important to acknowledge any limitations or challenges that were encountered during the study or the clinical case. This can provide a more balanced view of the results and their practical application.
Potential Future Research: The discussion could briefly touch upon potential avenues for future research or areas that require further investigation in the context of regenerative therapy for orthopedic pathologies in horses.
7. Conclusion: The text provides a general overview of the study and its findings, but there are some issues that need to be addressed:
Verb Tense Consistency: There is inconsistency in verb tenses throughout the text. For example, in the first sentence, it starts with "The study investigated," which is in the past tense. However, later in the same sentence, it switches to "MSC cells was capable," which is in the past tense but lacks subject-verb agreement. To maintain clarity, use consistent verb tenses throughout.
Clarification of Results: The text mentions that MSCs derived from adipose tissue were capable of differentiating into various cell types, but it doesn't specify the results of the experiments or their significance. Providing more details about the results and their implications would make the text more informative.
Clarity in Presentation: The text could be organized more clearly to separate the information related to the properties and capabilities of MSCs from the clinical case involving a horse with a subchondral cyst. Creating distinct sections for each aspect of the study would improve readability. References cannot be cited in conclusion.
8. generally, conclusion is too long
9. generally, Some sentences lack appropriate punctuation.
Author Response
We thank you for your efforts and valuable comments, which significantly contributed to increasing the quality of the manuscript. We have complied with all your requests which are highlighted in the red text by answering the questions you raised. The manuscript was in fact extensively modified according to your suggestions with the addition of some new references.
Round 2
Reviewer 1 Report
Comments and Suggestions for Authors
General Comments
This is a limited clinical study where subchondral bone cysts in a single horse joint was repaired using adipose derived MSCS mixed with platelet rich plasma.
Figure 1. The magnification is not 10x in version 1 and is not 4x in version 2. Also, the passage number of the cells should be included in the legend. The morphologic profile of these cells may be homogeneous, but this does not mean that the cellular population is physiologically homogeneous. The authors appear to be clueless about what determines magnification.
Image 3. Is this the image of the horse surgery? The image says that it is of a human subject.
The authors made no effort to address issues in the review.
Comments on the Quality of English LanguageEnglish language is OK, needs proofing.
Author Response
We have reviewed the photos and legends according to your indications.
Reviewer 2 Report
Comments and Suggestions for Authors
Thank you very much for the additional explanations added to the original manuskript. Unfortunately, they however do not change the fact that the conculsions drawn are not supported by the results.
Even though the application of MSC in a PRP-Gel seems to have led to a regeneration of the subchondral bone cyst in the reported case, there is no evidence, that the combination of the PRP-Gel with MSC was responsible for that effect. The same effect could have theoretically been achieved with MSC allone (as shown by Klein et al 2022) or with PRP-Gel allone. The lack of the respective control groups does not allow for the conclusion that the combination was crucial. Furthermore none of the results support, the statement that this "regenerative therapy method combined with the minimally invasive technique can offer an effective alternative for the treatment of other orthopedic pathologies in both animals and humans" as stated by the authors.
Since MSC were already shown to have a similar effect as other treatments (Klein et al 2022) and the case numebr of n=1 does not offer proof that the combination of the PRP-gel with MSCs was cruial to achieve the treatment effect the level of novelty of the findings in the current manuscript has to be considered very low.
Furthermore the authors discuss that the differentiation capacity of MSC is important for their regenerative potential. However, it is nowadays commonly agreed, that the treatment effect of MSC most likely relies on their paracrine effect and has nothing to do with a differentiation towards a tissue cell type of interest at the site of injury.
Vet Surg. 2022 Apr;51(3):455-463. doi: 10.1111/vsu.13782
Comparative results of 3 treatments for medial femoral condyle subchondral cystic lesions in Thoroughbred racehorses
Chelsea E Klein, Lawrence R Bramlage, Darko Stefanovski, Alan J Ruggles, Rolf M Embertson, Scott A Hopper
Comments on the Quality of English LanguageEnglish would need to be improved.
Author Response
Thanks to the reviewer’s suggestion, we have decided to remove the following sentence, as it is misleading: “The regenerative therapy method combined with the minimally invasive technique can offer an effective alternative for the treatment of other orthopedic pathologies in both animals and humans,” as stated by the authors.
In response to the reviewer's concerns regarding the work of Klein et al., we must point out that there are some differences that were overlooked in his/her comparison to our work:
The work of Klein et al 2022 describes 3 groups of horses with subchondral cysts treated not only with a) MSCs but also with b) arthroscopic debridement and c) corticosteroids. In that work, there are no differences in limb functional recovery in the 3 different treatments. Furthermore, neither instrumental analyses that demonstrate the regression of the lesion in relation to race prognosis nor follow-up radiographs demonstrate any actual production of new bone tissue.
We think our research may be of interest because it describes a specific clinical case with a critical-sized bone defect—2 cm in diameter, i.e. larger than those described in Klein et al. 2022—that had already undergone cyst enucleation treatments but did not experience functional limb recovery or bone tissue reconstruction. We used AMSCs trapped in the fibrin network for this desperate clinical case, which we obtained through the activation of thrombin, which is responsible for converting fibrinogen into fibrin during the coagulation of autologous PRP. In actuality, the fibrin scaffold, which had been locally coagulated, was perfectly adapted to the lesion site, offering a fundamental and transient three-dimensional structure capable of promoting cell migration, proliferation, and the formation of new bone tissue. We highlighted the synergistic action of AMSCs and PRP, which is essential for bone defect reconstruction, precisely because conventional methods for triggering bone tissue self-repair were insufficient.
We removed the sentence “The use of AMSCs with osteogenic potential, combined with platelet gel, could be the optimal solution for the repair of non-critical dimension bone defects” because the regenerative power of AMSCs is also due to their paracrine effect.